# Assessing the Adoption of One Health Approaches in National Plans to Combat Health Threats: The Pilot of a One Health Conceptual Framework in Armenia

**DOI:** 10.3390/tropicalmed9010022

**Published:** 2024-01-16

**Authors:** Alessia Milano, Claudia Robbiati, Silvia Declich, Paolo Calistri, Ombretta Pediconi, Laura Amato, Lusine Paronyan, Lilit Avetisyan, Arsen Manucharyan, Georgi Avetisyan, Tigran Yesayan, Arman Gevorgyan, Tigran Markosyan, Maria Grazia Dente

**Affiliations:** 1National Center for Global Health, Italian National Institute of Health (Istituto Superiore di Sanità), 00161 Rome, Italy; 2Department of Public Health and Infectious Diseases, Sapienza University of Rome, 00185 Rome, Italy; 3National Reference Centre for Veterinary Epidemiology, Programming, Information and Risk Analysis (COVEPI), Istituto Zooprofilattico Sperimentale dell’Abruzzo e del Molise ‘G. Caporale’, 64100 Teramo, Italy; 4Training and Project Management Unit, Istituto Zooprofilattico Sperimentale dell’Abruzzo e del Molise ‘G. Caporale’, 64100 Teramo, Italy; 5Zoonotic and Parasitic Diseases Epidemiology Department, National Center for Disease Control and Prevention, Yerevan 0096, Armenia; 6National Center for Disease Control and Prevention, Yerevan 0096, Armenia; 7Reference Laboratory Center Reference Laboratory of Episootology, Ectoparasitology and Entomology, National Center for Disease Control and Prevention, Yerevan 0096, Armenia; 8Veterinary Inspectorate, Food Safety Inspection Body, MoE, Yerevan 0010, Armenia; 9Risk Assessment Research Center, MoE, Yerevan 0071, Armenia

**Keywords:** One Health, Crimean–Congo hemorrhagic fever, anthrax, prevention, preparedness, Armenia, framework

## Abstract

Due to several factors, such as environmental and climate changes, the risk of health threats originating at the human–animal–environment interface, including vector-borne diseases (VBDs) and zoonoses, is increasing. Low-resource settings struggle to counter these multidimensional risks due to their already-strained health systems and are therefore disproportionally affected by the impact caused by these changes. Systemic approaches like One Health (OH) are sought to strengthen prevention and preparedness strategies by addressing the drivers of potential threats with a multidisciplinary and multisectoral approach, considering the whole system at the human–animal–environment interface. The integration of OH in national plans can be challenging due to the lack of effective coordination and collaboration among different sectors. To support the process of knowledge coproduction about the level of OH integration in prevention and preparedness strategies against health threats in Armenia, a situation analysis was performed on Crimean–Congo hemorrhagic fever/virus and anthrax (identified by local stakeholders as priorities to be addressed with the OH approach), and actions to strengthen the national OH system were identified with the support of a OH conceptual framework. The study highlighted that multidisciplinary and multisectoral efforts towards prevention and preparedness against VBDs and zoonoses threats need to be strengthened in Armenia, and priority actions to integrate the OH approach were identified.

## 1. Introduction

Evidence shows that environmental and climate changes impact the epidemiology of infectious diseases, especially those that involve a close interaction between humans, animals and the environment, like vector-borne diseases (VBDs) and zoonoses [1,2]. Low-resource settings are disproportionally affected by the impact caused by these changes, with the increased risk of VBDs and zoonoses hardly countered by strained health systems [3,4]. Therefore, enhancing prevention and preparedness strategies for VBDs and zoonoses is crucial to alleviate the burden of these threats and consequently improve health and socio-economic conditions [5].

As the recent pandemics of coronavirus disease (COVID-19), Middle East respiratory syndrome (MERS), Ebola, swine flu and severe acute respiratory syndrome (SARS) have highlighted, pathogens are sustained by multidimensional drivers. Systemic approaches, like One Health (OH), which are based on the collaboration of several sectors and stakeholders, can represent a valid strategy to identify common priorities and available resources and to understand and address complex interactions between disease and its multidimensional drivers [6,7,8,9,10]. International organizations have provided guidance to facilitate the integration of the OH approach in national strategies, and several studies have reported positive outcomes in addressing zoonoses and other infectious diseases with an OH approach [11,12,13,14,15,16,17,18,19,20,21,22,23].

There is a consensus on the need to strengthen prevention and preparedness strategies and to promote the institutionalization and implementation of national plans that include OH approaches by sensitizing and motivating the actors involved through capacity-building activities, such as assessments and exercises, aimed at identifying potential avenues for integration [24,25,26]. However, adopting OH in prevention, preparedness and response plans could be challenging given that the sectors involved usually work in silos. To move towards actual integration requires the investment of resources from different sectors to support multisectoral governance and capacity building and to make OH an integral part of national, regional and international strategies. A synergic OH prevention–preparedness system should be promoted rather than distinct OH action plans addressing specific sectors like zoonoses or antimicrobial resistance, especially when the available resources are limited [27]. In addition, the integration of OH in national prevention and preparedness strategies should identify locally tailored and sustainable actions that could valorize national strengths and opportunities.

This paper describes the assessment implemented with the Armenian National Institutions in the framework of the European Project MediLabSecure (MLS) [28]. The MLS project includes a network of 22 countries across the Sahel, Middle East, Balkans and Black Sea regions, and it aims to reinforce the OH prevention and preparedness capacities for VBDs, with a specific focus on arboviral diseases, a type of viral disease transmitted to humans through the bite of an infected arthropod.

Armenia is an upper-middle-income country; however, in 2021, 26.5% of people lived below the national poverty level, with peaks of 50% in fragile areas [29]. Climate change is already affecting Armenia, with an annual temperature increase higher than the global average and a significant decrease in precipitation [30]. The warming climate may also increase the risk of outbreaks of infectious diseases carried by ticks and mosquitoes, with people working in agriculture being at an increased risk [31]. This is particularly alarming given that agriculture is one of the key sectors of the Armenian economy [32]. In Armenia, as well as in many other former Soviet countries, the health care system was destabilized following independence in 1991, leading to a substantial decline in access to and quality of health services [33], and the situation has been exacerbated by the 2022–2023 Nagorno-Karabakh crisis [34,35]. There are high levels of out-of-pocket payments for healthcare, although there has been a steady increase in the coverage of health services since 2019 [36].

In terms of OH adoption in national strategies, a relevant effort has been implemented towards the integration of the human and animal health sectors by assessing capacities and opportunities to enhance the OH national system [37,38,39].

However, synergies with other relevant sectors and other national strategies, like the National Action Program of Adaptation to Climate Change (2021–2025), are not yet adopted [30].

With the aim of contributing to the strengthening of the Armenian OH system with the identification of areas to be addressed and priority actions to implement, we explored the extent, successes and challenges of OH integration in prevention and preparedness strategies to VBDs and zoonotic diseases.

## 2. Materials and Methods

### 2.1. The One Health Conceptual Framework (OHCF)

We adopted the One Health conceptual framework (OHCF) [40], elaborated in the context of the Group of Twenty (G20) 2021 meeting [41], to understand to what extent the OH approach is included in national strategies for prevention and preparedness against VBDs and zoonoses in Armenia.

The OHCF aims at strengthening the OH national systems by highlighting gaps in existing prevention, preparedness and response strategies and by identifying national procedures that support the integration of OH approaches. The framework identifies five national targets and related priority actions that should be considered to ensure OH operationalization and implementation, such as governance mechanisms, prevention and preparedness plans, data collection systems, capacity-building activities and evaluation strategies (Table 1).

The national targets should be ensured by the individual countries, considering the situation at a national level and the specific constraints and opportunities, with the support of guidelines and tools produced at the international level by the relevant organizations.

The framework also identifies two international targets aimed at guiding the development of national OH plans on the basis of harmonized global strategies elaborated by relevant organizations, including the Quadripartite (international framework established by the World Health Organization, the World Organization for Animal Health, the Food and Agriculture Organization of United Nations and the United Nations Environment Program to facilitate OH operationalization).

### 2.2. Study Design

The study has been developed and guided by a team comprising investigators from the Italian National Institute of Health (Istituto Superiore di Sanità, ISS), Istituto Zooprofilattico dell’Abruzzo e del Molise, Giuseppe Caporale (IZS—Teramo) and representatives of the Armenian National Centre for Disease Control and Prevention (NCDC) and the Ministry of Economy of Armenia (in 2021, the Ministry of Agriculture merged with the Ministry of Economy).

This study was a situational analysis with the aim of coproducing knowledge about the level of integration of the different sectors involved in prevention and preparedness strategies for VBDs and zoonotic diseases in Armenia, based on a methodology developed within the MLS project [42]. The OHCF was introduced in this study as a tool to identify strengths and gaps in OH operationalization and implementation.

We adapted the methodology to be fully virtual, considering the restrictions during the COVID-19 pandemic. Three workshops, each lasting around three hours, were implemented online through the Cisco Webex^®^ platform for meetings and webinars and recorded to allow the retrieval of missing information. The study team was facilitating information sharing and discussion during the workshops.

The study started in June 2021 and ended in July 2022 and engaged relevant stakeholders from different sectors in Armenia. The Armenian context referents from the NCDC and the Ministry of Economy were engaged from the planning phase of the study, and they promoted and facilitated the involvement of all the other relevant institutions.

We identified four specific objectives for the study:Describe how the collection, analysis and dissemination of information is organized within and between relevant sectors involved in the surveillance systems.Discuss the main challenges and enabling factors in establishing a functional inter-sectoral utilization of the information collected across sectors.Assess gaps and opportunities in the adoption of OH approaches in prevention and preparedness strategies with the support of the OHCF.Identify key actions to enhance OH integration and assess their feasibility.

The study was implemented through the progressive phases reported in Table 2, which started with a preliminary picture of the situation and terminated with the identification of priority actions to be implemented to reinforce the OH national system.

## 3. Results

### 3.1. Preparatory Phase

Key national stakeholders were consulted to reach a consensus on the objectives and the methodology of the study. A preliminary country study portfolio was developed in advance through a review of the available documentation and consultation with the Armenian stakeholders. The portfolio, developed and used also in previous studies in MLS [42], included a checklist to assess and document the level of integration between sectors and the involvement and role of each of them. It helped in identifying information gaps and facilitating discussions during the workshops about lessons learned and barriers affecting the integration between different sectors (animal, human, vector and environment).

During the consultation with the Armenian stakeholders, they provided a list of priority pathogens that would benefit from the integration of OH in their prevention, preparedness, surveillance and response. It was suggested to include in the study at least an arboviral threat in line with the scope of the MLS Project (Table 3).

A stakeholder mapping was performed and included representatives of the Armenian institutions from different sectors and country experts who took part in previous studies/exercises at a national level (i.e., the Joint External Evaluation and the National Bridging Workshop) [37,38]. A total of 19 representatives from 12 Armenian institutions took part in the study, as reported in Table 4.

Some additional experts took part in the workshops if invited by representatives of the institutions.

With reference to the prioritization of the zoonotic and vector-borne pathogens, the criteria for ranking were identified on the basis of the guidance of WHO [43] and the objectives of the study.

The following criteria were considered:-Emerging or re-emerging threat with a changing of pattern distribution.-Emerging or re-emerging threat at the human–animal–environment interface requiring a multisectoral action (OH approach).-Available surveillance system or/and plans or/and recent response actions to the threat.

Based on the above criteria, the following indicators were proposed by the study team for the prioritization of the pathogens to be agreed upon during the first workshop:Threat which has activated a recent response action to contain a potential outbreak of the disease.Threat for which an OH preparedness/surveillance plan is available.Threat detected or caused outbreaks/epidemics in the past ten years.Threat affecting food safety and/or food security.Threat benefitting the most from the integration of environmental and climatic data in its/their surveillance.Threat benefitting the most from the integration of OH in preparedness/surveillance/response.Threat greatly impacting socio-economic aspects in case of an outbreak.Threat detected in a new location or population in the country or neighbouring countries in the past ten years.Threat whose animal host (domestic or wild) is/are in close proximity to humans.Threat whose related vector/s’ presence and abundance are increasing due to anthropogenic, climatic and environmental factors.Threat with an integrated data collection and analysis system.

### 3.2. The First Workshop: Prioritization of Pathogens

The aim of the first workshop was to prioritize the zoonotic and vector-borne pathogens previously identified (Table 3).

The workshop was organized with a participatory approach, according to the following steps:Seeking consensus about criteria and indicators for the prioritization of the selected pathogens;Ranking the pathogens on the basis of the agreed prioritization indicators.

The pathogens were prioritized using the Cisco Webex^®^ platform for meetings and webinars and the Slido^®^ application for managing live polls in the Webex platform through two virtual polling sessions. In the first poll, the arboviral pathogens were prioritized and Crimean–Congo hemorrhagic fever virus (CCHFV) was the pathogen that received the highest score (35), followed by West Nile virus (WNV) (26) and Rift Valley virus (RVFV) (13) (Table 5).

In the second poll, which focused on the other pathogens, anthrax was the one that received the highest score (39), followed by brucellosis (38), leishmaniosis (21), rabies (14), tick-borne encephalitis (TBE) (9) and dirofilariasis (8) (Table 6).

CCHFV and anthrax were deemed to benefit the most from the adoption of the OH approach in prevention, preparedness, surveillance and response strategies. In particular, the surveillance of CCHFV could be enhanced by the integration of environmental and climatic data, given that the circulation of the pathogen is connected to anthropogenic, climatic and environmental factors. Prevention of anthrax can be strengthened by vaccination and appropriate land use, which implies a multisectoral effort. It is reported that the occurrence of anthrax outbreaks is particularly associated with the inappropriate use of land, as it happens with seasonal grazing when the density of livestock on pasture influences the incidence of disease [44]. Moreover, CCHFV outbreaks in neighboring countries and past and recent anthrax epidemics are threatening the Armenian health and socio-economic systems [45].

### 3.3. The Second Workshop: Framing the OH System

#### 3.3.1. The Surveillance Systems

The surveillance systems of the two selected pathogens were described and discussed during the second workshop with the Armenian stakeholders with the main aim of identifying possible integration pathways among sectors.

(1)CCHFV surveillance system in Armenia

CCHFV is a zoonotic viral infection that can manifest with a range of symptoms from influenza-like to hemorrhagic syndromes and a mortality rate ranging from 5% to 50% [46]. Human infections might occur through different routes including direct tick bite and direct contact with blood and other tissues of infected animals or humans. CCHFV is a member of the *Bunyaviridae* virus family. The virus has been isolated from at least 31 species of ticks belonging to the *Ixodidae* (hard ticks) and *Argasidae* (soft ticks) families [46].

The first detection of CCHFV in ticks and the only laboratory-confirmed human case in Armenia dates back to 1974 [46,47], and there are no published reports of CCHFV activity in Armenia in the last five decades [46]. In spite of the recent worsening of the CCHFV situation in neighboring Turkey and Iran, there is no consistent monitoring of CCHFV key vector distribution and abundance in Armenia [46]. The available data suggest that Armenia is considered a country where CCHFV cases are reported intermittently in the absence of robust surveillance (Figure 1) [47]. A more recent paper detected CCHFV antigens in tick samples for the first time in the last five decades [48].

A case-based surveillance system for human cases of CCHF, whose effectiveness until now has never been evaluated, is in place in Armenia under the NCDC. CCHFV is on the list of immediate notification, and its diagnosis is in accordance with EU directives [49]. An indicator-based surveillance system and a syndromic surveillance program for hemorrhagic fevers and fevers with rash cases are in place (information provided during the second workshop). No veterinary surveillance system is in place for CCHFV. The Reference Laboratory Center of the Food Safety Inspection Body remains the main reference institution for diagnostics. The center mainly performs activities to adhere to the World Organisation of Animal Health (WOAH) policies (information provided during the second workshop). Equipment to perform diagnostic tests is generally available, including a small quantity of kits for CCHFV. Medical entomologists collect *Ixodes* ticks and create maps of distribution in the country. Data about possible outbreaks are shared with the veterinary services for early detection and prevention. Cross-border collaborations are regulated by formal frameworks, but they are not always implemented. This is the case of the last outbreak of CCHFV in Iran in 2018, when collaboration between the two countries was not activated although a formal framework for collaboration was in place (information provided during the third workshop).

(2)Anthrax surveillance system in Armenia

Anthrax is an infectious disease caused by a Gram-positive bacteria known as *Bacillus anthracis*. It commonly affects domestic and wild animals, but people can become sick with anthrax if they come in contact with infected animals or contaminated animal products. Anthrax can cause rapidly large animal losses [50]. Armenia is at high risk for anthrax, both at animal and human levels, as the country is characterized by a number of permanent at-risk areas. Their presence is probably due to climatic factors and various characteristics of the landscape, soil and vegetation cover, as well as the intensity of livestock production.

During the period from 2000 to 2015, 20 cutaneous anthrax cases were detected [51,52]. In August 2019, two outbreaks of anthrax occurred in Armenia [53,54]: the first happened in the Gegharkunik region, and the second one took place in the Armavir region [55]. The most recent outbreaks took place in 2021 in the Gegharkunik and Shirak regions. Its management was discussed thoroughly during the second workshop to assess to what extent the OH approach was adopted. The WHO International Health Regulations (IHR-2005) national committee was not activated during this outbreak because it was not considered a national emergency under the WHO IHR-2005 legal framework.

A case-based human surveillance system is in place, but the Electronic Diseases Surveillance System (EDSS) for human and animal infectious diseases is not yet operative, so the data collected cannot be shared between sectors electronically. At the animal level, surveillance activities are not implemented, and preventive efforts are focused on vaccination. Each year, under the “Agricultural Livestock Vaccination Program” of the Ministry of Agriculture (in 2021, it merged with the Ministry of Economy), the vaccination of all the registered cattle population (every six months) and small ruminants (once a year) is implemented. Also, the veterinary services are supported by 685 community veterinarians, specialists who graduated from a veterinary higher education institution, who inform the regional offices of government in case of suspected symptoms. They are mainly veterinarians who work in veterinary services under the Ministry of Economy. People with other backgrounds can be community veterinarians, but they can only perform vaccinations (information provided during the second workshop).

#### 3.3.2. The OH Assessment through the OHCF

The second part of the workshop was dedicated to the assessment of actions in place and the areas that need to be addressed (Table 7) to ensure the integration of OH approaches in the prevention, surveillance and preparedness for CCHFV and anthrax in Armenia with the support of the OHCF [36].

### 3.4. The Third Workshop: Strengthening the OH System

The assessment conducted during the second workshop highlighted the areas to be addressed to enhance the OH system in Armenia (Table 7). During the third workshop, the opportunities available to address the areas in need of CCHFV and anthrax prevention and preparedness and to enhance the integration of the system were discussed.

Areas for possible improvement emerged from the analysis, and their feasibility was assessed with the Armenian stakeholders. A scoring system from 1 (not feasible) to 5 (feasible) was adopted, and the opportunities that scored 4 (80% chance of feasibility) or more were considered feasible with minor barriers.

Regarding CCHFV opportunities for OH integration, the stakeholders judged the development of a preparedness plan for arboviral diseases with an OH approach to be feasible. Serological surveys on domestic animals (especially ruminants) to define areas exposed to the virus and the surveillance of tick infestations in domestic animals in at-risk areas to improve early warning activities were described as feasible. Distribution maps of ticks within the country and the monitoring of hotspots were also deemed feasible on the basis of the activities in progress in this field. Moreover, to improve risk assessment strategies, serological surveys on at-risk groups could be performed to identify areas exposed to the virus. Finally, integrated data collection and analysis to monitor at-risk groups, at-risk areas and at-risk events were described as feasible (Table 8).

For anthrax, two opportunities were judged to be feasible (score 4 or more), namely performing studies to explore anthrax drivers in order to guide prevention actions and enhancing the engagement of communitarian vets in preparedness and prevention actions (including vaccination) (Table 9).

Opportunities to enhance intersectorality and strengthen the national OH system were deemed more difficult in the short term because they require substantial changes in the governance structure and adequate resource allocation (Table 10).

## 4. Discussion

The distribution of VBDs, zoonoses and other infectious diseases is increasing due to global changes, including those related to the environment and climate. Epidemics and epizootics affect countries’ development in several ways, by impacting human health, animal health, the healthcare system, the local economy and access to education [1,2,3].

A multidisciplinary and multi-stakeholder effort towards prevention and preparedness for these threats is needed to reduce the impact on national health and socio-economic systems and the exacerbation of inequality and poverty [4]. OH can support this if its integration in preparedness and prevention strategies is operationalized and implemented. Toward this aim, assessments and exercises conducted nationally with multidisciplinary stakeholders can increase the knowledge and awareness of the institutions involved regarding strategies and procedures that could enhance the adoption of multisectoral approaches.

This study has assessed surveillance and preparedness for CCHFV and anthrax in Armenia and areas for improvements have been identified and discussed to consolidate the OH integration in Armenian national strategies.

The methodology adopted to prioritize the threats to be assessed has introduced the concept of OH threat (emerging or re-emerging pathogen at the human–animal–environment interface requiring multisectoral OH action). As a consequence, multisectoral criteria and indicators for prioritization have been developed.

The involvement of national stakeholders during all steps of the study and the use of the OHCF allowed the identification of appropriate and feasible avenues to capacitate the local OH system. The stakeholder mapping led by local contact points supported the inclusion of all the relevant actors. Moreover, it highlighted that the environmental and socio-economic sectors seem to be rarely involved in the actual development of prevention and preparedness actions, despite their crucial role in VBD and zoonotic outbreaks and the socio-economic consequences.

The assessment was also an opportunity for the representatives of the veterinary services to share with the other sectors their concerns regarding the recurrent anthrax outbreaks and the need to find common resources and procedures to ensure the accomplishment of all the activities needed to prevent further outbreaks (i.e., early detection of cases among animals, proper removal of carcasses and complete burning or burying of dead animals, keeping dogs and other animals out of places where the animals were buried, supporting decontamination activities of the territory, ensuring compliance with the prohibition of slaughtering of dead animals and strengthening measures against the uncontrolled movement of animals and meat products). Also, the efficacy of vaccination campaigns might be compromised by the fact that the animal registration system does not cover all the animals circulating in the country.

The study has facilitated the discussion around other aspects related to OH integration, like the need for improving information sharing both at a national and regional level, ensuring real-time data exchange and joint risk assessments. Assessing the level of integration of the different sectors in the surveillance of pathogens has called attention to the need for a national common data system that could support an integrated early warning system.

One aspect to be better considered in Armenia, like in other countries, is the well-established WHO IHR-2005 capacity. During the outbreak of anthrax in Armenia in 2021, this capacity was not exploited in full because it was not considered relevant to WHO-IHR requirements (i.e., it was not an international outbreak). The valorization of national WHO IHR-2005 capacity could enhance OH approaches and support effective prevention and preparedness measures, also with bordering countries [8,15]. As an example, CCHFV is circulating in all the countries bordering Armenia, and the assessment has confirmed that some feasible opportunities are available to monitor the actual virus’s circulation in the area. However, the lack of an integrated data system is a barrier preventing joint risk assessments from being performed in collaboration with neighboring countries.

This study has some limitations. The assessment was entirely conducted remotely, with some challenges for the participation of all the stakeholders. Data collection during similar previous studies within the MLS project was conducted during country visits. The fact that these Armenian institutions have been part of the MLS project since 2014 increased their awareness about the OH approach, and stakeholder participation in the workshops was constant and active. However, even if conducted remotely, the study was an opportunity for sharing information and documentation between the stakeholders involved, and it possibly increased their awareness about the need for a multisectoral approach.

## 5. Conclusions

Through the adoption of OH in prevention and preparedness strategies, countries can gain a better understanding of drivers linked to the emergence of pathogens or recurring outbreaks, which cause impoverishment of national resources and hamper sustainable development. Situation analyses conducted with a multidisciplinary approach increase the awareness of the institutional stakeholders about potential areas benefitting from the integration of OH approaches.

There is also the need to constantly repeat these assessments to monitor progress and highlight persistent barriers through piloted methodologies and tools.

The integrated surveillance and the Electronic Diseases Surveillance System, which were considered priorities for Armenia both in the outcome of the WHO Joint External Evaluation of IHR Core Capacities in 2016 [37] and in the National Bridging Workshop on IHR and PVS Pathway in 2019 [38], were still not operational at the time of this study. Sharing data between sectors with electronic systems is one of the main barriers to OH operationalization, affecting multisectoral real-time surveillance and risk assessments in many countries [40]. The need to include additional relevant sectors, such as environment, socio-economic and communities, in the development and implementation of prevention and preparedness plans with an OH perspective is recognized but not yet fully implemented in Armenia, as in many other countries.

Tools like the OHCF could help to systematically assess the level of OH integration within national systems and identify barriers and enablers to be considered also in a synergistic effort with the frameworks provided by the relevant international organizations.

## Figures and Tables

**Figure 1 tropicalmed-09-00022-f001:**
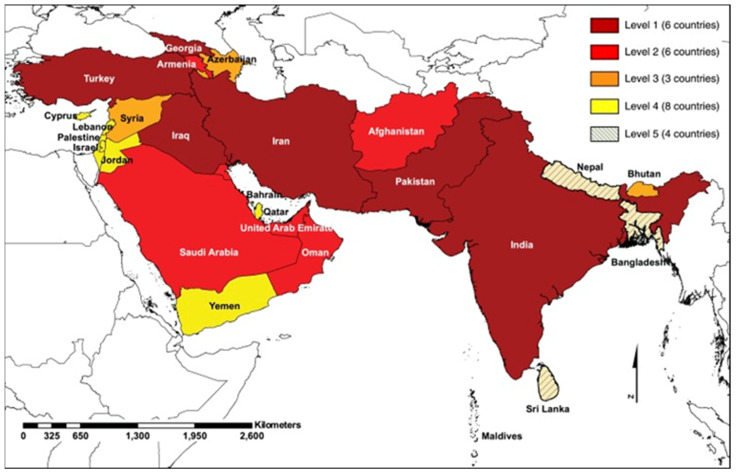
Burden of CCHFV in Southern and Western Asia [47].

**Table 1 tropicalmed-09-00022-t001:** One Health conceptual framework (OHCF) [40].

Scope: to Facilitate OH Operationalization and Implementation for the Prevention and Preparedness for Threats at the Human–Animal–Environment Interface.						
Targets						
National Level					International Level	
Target 1	Target 2	Target 3	Target 4	Target 5	Target 1	Target 2
governance	prevention and preparedness	data collection and analysis	capacity building	consolidation and evaluation of the system in place	harmonisation of plans and cross-border collaborations	harmonised capacity building
National legislative and procedural framework that allows/imposes mainstream OH approaches in all the prevention and preparedness strategies and allocates the necessary resources. One Health national centers.	Prevention and preparedness plans developed, implemented and monitored with an OH approach, including community empowerment strategies, for the prevention and containment of health threats	National sector-driven database interoperable and accessible to all the institutions/sectors involved in the One Health team	National training plan on OH strategies agreed upon between institutions and integrated into the relevant national plans	Evaluation plans to assess the effectiveness of OH in reducing the risks of potential epidemics (prevention); supporting the early identification of epidemic risks (alerting); and contributing to a reduction in impact (mitigation)	International Framework enabling harmonized integration of OH strategies in all relevant regulations and communications	International training plans and tools aimed at facilitating OH training at national and cross-country levels
**Priorities for Action**						
**National level**					**International level**	
**Priority 1**	**Priority 2**	**Priority 3**	**Priority 4**	**Priority 5**	**Priority 1**	**Priority 2**
governance	prevention and preparedness	data collection and analysis	capacity building	consolidation and evaluation of the system in place	harmonisation of plans and cross-border collaborations	harmonised capacity building
Establishing a national multisectoral and multistakeholder team to set principles, rules and procedures to allow operationalization of OH strategies Assessing the opportunity and benefits of setting up an OH national center Enacting laws and identifying resources for OH operationalization	Connecting OH strategies to prevention and preparedness plans by establishing a multisectoral team (OH team) in charge of the development, implementation and monitoring of plans	Identification of national priority areas to be monitored and related monitoring indicators/metrics Verifying available sources of information and data Development of an integrated and interoperable database connected with early warning and surveillance systems	Development of training curricula on OH prevention and preparedness Training of staff involved in activities including OH strategies Piloting and exercising	Identifying monitoring and impact indicators Assessing level of implementation of OH indications in prevention and pandemic plans Assessing added value of OH in prevention and preparedness	Developing and updating guidance and regulations to integrate OH strategies in prevention and preparedness plans and international early warning systems Identification of OH preparedness indicators/metrics in collaboration with national OH teams Establishing Quadripartite-collaborating centers at national OH centers Facilitating networking opportunities between OH national centers	Integration of OH principles in international training for preparedness and in tools aimed at assessing the level of capacities Promoting harmonized and multicounty exercises

**Table 2 tropicalmed-09-00022-t002:** Phases of the study.

**Preparatory phase**—Involved stakeholders: Armenian context experts from the NCDC and the Ministry of Economy	
**Methods/Tools**	**Outcomes**
Desk review	Preliminary picture of the OH system in Armenia
	Analysis of outcomes provided by previous assessments (e.g., WHO JEE of IHR Core Capacities in 2016 and the National Bridging Workshop on IHR and PVS Pathway in 2019)
	Stakeholders mapping
Consultation with Armenian context experts	List of priority zoonotic and vector-borne pathogens
	Identification of criteria and indicators for the prioritization
**Workshops**—Involved stakeholders: Experts from all the institutions identified with the stakeholder mapping	
**Methods/Tools**	**Outcomes**
**1st workshop: prioritization**	
Consensus on criteria and indicators for the prioritization of zoonotic and vector-borne pathogens	Priority pathogens to be addressed during the study
Prioritization as per set criteria and indicators	
**2nd workshop: framing the OH system**	
Description of the surveillance systems for the selected pathogens	Status of integration between sectors
Analysis of the integration of OH approaches in prevention, surveillance, and preparedness plans of the selected pathogens through the OHCF	Strategies and procedures in place and areas of improvement
**3rd workshop: strengthening the OH system**	
Discussion about the available opportunities and their feasibility for the enhancement of the OH system in Armenia	Priority actions to be implemented and their perceived feasibility

**Table 3 tropicalmed-09-00022-t003:** List of pathogens for the prioritization.

Arbovirus (Focus of the MLS Project)	Other Identified Pathogens		
Virus	Virus	Parasites	Bacteria
Crimean–Congo hemorrhagic fever	Tick-borne encephalitis	Leishmaniosis	Anthrax
West Nile fever			
Rift Valley fever	Rabies	Dirofilariasis	Brucellosis

**Table 4 tropicalmed-09-00022-t004:** List of Armenian institutions involved in the study.

# of Representatives	Institution
1	National Center for Disease Control and Prevention/Ministry of Health—Direction
1	National Center for Disease Control and Prevention/Ministry of Health—Zoonotic and Parasitic Diseases Epidemiology Department
2	National Center for Disease Control and Prevention/Ministry of Health—Reference Virology Laboratory
1	National Center for Disease Control and Prevention/Ministry of Health—Laboratory of Episootology, Ectoparasitology and Entomology
1	Regional branch of National Center for Disease Control and Prevention in Shirak Province
6	Veterinary Inspectorate, Food Safety Inspection/Ministry of Economy
1	Risk Assessment Research Center/Ministry of Economy
1	Reference Laboratory of Especially Dangerous Pathogens, Republican Veterinary and Phytosanitary Laboratory Services Center
1	Incident Management Support Officer, WHO Armenia Country Office
2	CH2M [Engineering company named from the initials of its four founders. In 2017, CH2M became part of Jacobs Engineering Group Inc. (Welcome to Jacobs|Jacobs)]—Jacobs Armenia Branch Office
1	Center for Ecological-Noosphere Studies, National Academy of Sciences
1	Ministry of Environment

**Table 5 tropicalmed-09-00022-t005:** Pool results of the arbovirus pathogens (MLS project).

Questions	CCHFV	RVFV	WNV	Do Not Know	Total
Select pathogen/s which have activated a recent response action to contain a potential outbreak of this disease	0	0	0	4	4
Select the pathogen/s for which an OH preparedness/surveillance plan is available in Armenia	0	0	0	5	5
Select the pathogen/s which have been detected or caused outbreaks/epidemics in the past 10 years in Armenia	0	0	0	0	0
Select the pathogen/s that can affect food safety or/and food security	3	1	4	3	11
Select the pathogen/s which can benefit the most from the integration of environmental and climatic data in its surveillance	3	2	4	1	10
Select the pathogen/s which can benefit the most from integration of OH approach in preparedness/surveillance/response in Armenia	7	1	4	1	13
Select the pathogen/s which can have a big impact on economic and social aspects in case of outbreak in Armenia	7	5	6	1	19
Select the pathogen/s that have been detected in a new location or population (human or animal) in the country or neighboring countries in the past 10 years	5	0	3	0	8
Select the pathogen/s whose animal host (domestic or wild) is in close proximity to humans in Armenia	4	2	2	2	10
Select the pathogen/s whose related vector/s’ presence and abundance are increasing in Armenia due to anthropogenic, climatic and environmental factors	5	1	2	3	11
Select the pathogen/s with an integrated (human, veterinarian, environmental) data collection and analysis system in Armenia	1	1	1	2	5
Total	35	13	26	22	96

**Table 6 tropicalmed-09-00022-t006:** Pool results on other OH threats.

Questions	Anthrax	Brucellosis	Dirofilariosis	Leishmaniosis	Rabies	TBE	Do Not Know	Total
Select pathogen/s which have activated a recent response action to contain a potential outbreak of this disease	5	3	0	1	1	0	0	10
Select the pathogen/s for which an OH preparedness/surveillance plan is available in Armenia	1	2	0	1	1	1	2	8
Select the pathogen/s which have been detected or caused outbreaks/epidemics in the past 10 years in Armenia	3	3	1	1	1	0	1	10
Select the pathogen/s that can affect food safety or/and food security	8	8	0	0	1	1	0	18
Select the pathogen/s which can benefit the most from the integration of environmental and climatic data in its surveillance	2	1	2	3	0	2	0	10
Select the pathogen/s that can benefit the most from integration of OH approach in preparedness/surveillance/response in Armenia	6	6	2	3	3	1	0	21
Select the pathogen/s which can have a big impact on economic and social aspects in case of outbreak in Armenia	6	6	0	0	2	1	0	15
Select the pathogen/s that have been detected in a new location or population (human or animal) in the country or neighboring countries in the past 10 years	2	1	1	3	1	1	1	10
Select the pathogen/s whose animal host (domestic or wild) is in close proximity to humans in Armenia	5	5		4	4	1	1	20
Select the pathogen/s whose related vector/s’ presence and abundance are increasing in Armenia due to anthropogenic, climatic and environmental factors	1	3	2	5	0	1	0	12
Select the pathogen/s with an integrated (human, veterinarian, environmental) data collection and analysis system in Armenia	0	0	0	0	0	0	2	2
Total	39	38	8	21	14	9	7	136

**Table 7 tropicalmed-09-00022-t007:** Actions for OH consolidation in Armenia.

Identified Actions towards OH Integration in Armenia		
Targets as Per OHCF	Areas to Be Addressed	Actions and Procedures in Place
Governance	National procedures and mechanisms enabling intersectoral activities are not fully in place.	Under the International Health Regulations (IHR-2005), there are two multisectoral groups: 1. The high-level intersectoral steering committee. 2. The expert group under the MoH. Under the guidance of WHO, NCDC is developing an OH framework for the implementation of the tripartite guide, starting with legislation aspects.
Prevention and preparedness	Specific preparedness plans for pathogens are not available, only a generic preparedness one. Prevention and preparedness plans with an OH approach were reportedly under development but are not yet available.	After a WHO workshop (March 2022) on the control and prevention of zoonotic diseases, an OH national plan was developed and its operationalization was under discussion among the relevant stakeholders during the assessment.
Data collection and analysis	Data are not shared regularly between sectors.	The Electronic Diseases Surveillance System (EDSS) for human and animal infectious diseases has been developed but not operationalized at the time of the assessment.
Capacity building	A specific intersectoral training curriculum on OH is not available.	In 2021, training was organized by the NCDC, with WHO support, for different disciplines and sectors. US Defense Threat Reduction Agency (DTRA) supports frontline field epidemiology training with epidemiologists from public and animal health. In March 2022, the WHO country office implemented a training workshop focused on the operationalization of the tripartite action plan.
Consolidation and evaluation of the system in place	Specific evaluation/s of the national OH system are not available.	Assessments of collaboration capacities were implemented: the WHO Joint External Evaluation in 2016 [34], the National Bridging Workshop on the International Health Regulations (IHR) and the WOAH Performance of Veterinary Services (PVS) Pathway in 2019 [35].
National actions supporting international harmonization and cross-border collaboration	Cross-border collaborations.	MoH has a Memorandum of Understanding with Iran and Georgia for information sharing.

**Table 8 tropicalmed-09-00022-t008:** Opportunities to enhance CCHFV prevention and preparedness.

Areas of Improvement	CCHFV Prevention and Preparedness	Feasibility Score
Human and animal surveillance	Cross-sectional serological surveys on domestic animals (especially ruminants) to define areas exposed to virus infections	4.5
	Surveillance of tick infestations in domestic animals in at-risk areas	4
Vector mapping	Consolidated maps of the distribution of ticks in the country and identification of priority areas to be monitored	4.5
Risk assessment	Serological surveys on at-risk groups to define areas exposed to the virus	4.8
	Collection and analysis of data to monitor at-risk groups, areas and events	4.3

**Table 9 tropicalmed-09-00022-t009:** Opportunities to enhance anthrax prevention and preparedness.

Areas of Improvement	Anthrax Prevention and Preparedness	Feasibility Score
Identification of drivers	Studies on drivers of anthrax in order to guide prevention actions	4.5
Community engagement	Communitarian vets more involved in preparedness and prevention actions	4.5

**Table 10 tropicalmed-09-00022-t010:** Opportunities to enhance the One Health system.

	Intersectorality	Feasibility Score
Governance	Leveraging on WHO IHR-2005 requirements to enhance intersectoral activities at national level	3.8
	Preparedness plans with an OH approach	4
	Multisectoral training curriculum on One Health	3.8
	OH operationalization (enactment of national laws and procedures)	3.8
	Include climate and environmental data and identify trends and drivers	3.8
	Prioritize training on the Electronic Diseases Surveillance System (EDSS) for all the sectors involved at national and peripheral levels	3.8
	Education and awareness campaigns for communities involving all sectors	3.3

## Data Availability

All data and material used in the framework of this study are available on request.

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
