# Peer review of "Assessing the Adoption of One Health Approaches in National Plans to Combat Health Threats: The Pilot of a One Health Conceptual Framework in Armenia"

_tropicalmed, 2024, doi:10.3390/tropicalmed9010022_

Round 1

Reviewer 1 Report

Comments and Suggestions for Authors

Dear Authors,

This work is well researched, presented, developed, and cited. Please see my comment items in attached pdf. In short, appreciate the narrative , top-down approach on One Health, and background related to VBD and zoonoses in Armenia.

Related to materials/methods and results section, can you consider reinforcing your Fig 1 to illustrate-high level and otherwise map of your overall activity? It would be important to show process and from there you can further draw areas where you learned need further attention. Otherwise, the work is well documented but perhaps helping a reader through the process would make it more impactful rather than seeing your observations and results.

Also suggest you reconcile methods/materials/results and supplemental material. The agenda items are nice but as supplemental material they do not add a lot or stand alone.

Finally, consider what take away messages can be drawn in your conclusion section? I assume there are observations you experienced-so what was similar (for example) with Armenia and other countries of similar size/income, etc and what was dissimilar? Perhaps you will draw some interesting points to further draw readers from different backgrounds and locations.

Best of luck, appreciate the opportunity to review this work.

Comments on the Quality of English Language

This work was written and presented well. Minor items with English-terms and understand this may be editorial discretion.

Author Response

Dear Authors,

This work is well researched, presented, developed, and cited. Please see my comment items in attached pdf. In short, appreciate the narrative , top-down approach on One Health, and background related to VBD and zoonoses in Armenia.

Related to materials/methods and results section, can you consider reinforcing your Fig 1 to illustrate-high level and otherwise map of your overall activity? It would be important to show process and from there you can further draw areas where you learned need further attention. Otherwise, the work is well documented but perhaps helping a reader through the process would make it more impactful rather than seeing your observations and results.

Also suggest you reconcile methods/materials/results and supplemental material. The agenda items are nice but as supplemental material they do not add a lot or stand alone.

Finally, consider what take away messages can be drawn in your conclusion section? I assume there are observations you experienced-so what was similar (for example) with Armenia and other countries of similar size/income, etc and what was dissimilar? Perhaps you will draw some interesting points to further draw readers from different backgrounds and locations.

Best of luck, appreciate the opportunity to review this work.

Comments on the Quality of English Language

This work was written and presented well. Minor items with English-terms and understand this may be editorial discretion.

Submission Date

06 December 2023

Date of this review

22 Dec 2023 20:41:32

Authors’ reply

Dear reviewer,

thanks a lot for your comments which helped us to proceed with a comprehensive revision of the manuscript.

We tried to addressed all your comment items as following:

Line 38 in the revised manuscript (previous line 38): amended as suggested

Line 81 (previous 81): spell typo amended

Lines 92&97 (previous 82): references considered and integrated

Lines 101-104 (previous 93-100): paragraph amended as suggested

Lines 118-126 (previous 109): we added a paragraph to explain what constitutes a national vs international target.

Line 152 (previous 129): amended as suggested

Lines 146-171 and Table 2. (previous line 132 and Figure 1.): We revised entirely the figure 1. and we transformed it in Table 2 (in our view more explicative than a graphic). With this we addressed your valuable points related to showing the adopted process,  reconciling methods/materials/results and avoiding supplemental material.

Lines 216-243 (previous 46-75 pag7 under results): re-organised as suggested

Lines 273-279 (previous 89-94 pag 10 under results): integrated with details and reference as suggested

Line 294 (previous 108): we added related reference.

Line 336-338 (previous 147 pag 12 under results): Electronic Diseases Surveillance System (EDSS) for human and animal infectious diseases not yet operative. Detail included in the revised text.

Table 7 (previous Table 6-Capacity building): information provided during the Workshops. No public references available.

Line 421 (previous 236): typo checked

Line 434 (previous 248): The references [37] and [38] are tools to monitor IHR national capacities. We deem that discussing this aspect is out of the scope of our study.

Line 451 (previous 266-): Conclusions: we integrated the conclusions as per your suggestions.

Kindly find in attachment the revised manuscript as per your suggestions

Reviewer 2 Report

Comments and Suggestions for Authors

The authors have worked under difficult situations, i.e. Covid-19 pandemic and they tried to do their best to prepare this manuscript. The importance of OH and challenges in One Health integration of different sectors for prevention and preparedness for VBDs and zoonoses has been elaborated in the Armenian context. The proposed One Health Conceptual Framework is based on the impact of climate change on the emergence of vector-borne diseases of zoonotic origin. The authors have recognized the limitations of the study. It will be good to reflect on the Joint External Evaluation report and recommendations to be linked to strengthening IHR capacities such as preparedness, risk assessment, response, and multisectoral coordination  (One Health) and surveillance rather than IHR (2005) itself.

Stakeholder mapping (not analysis) to analyze roles and responsibilities of different sectors in prevention and preparedness for VBDs and zoonoses. MLS project was designed to understand the One Health approach for vector-borne diseases such as CCHF, RVF, and WNV in the context of climate change in Africa, and Armenia was also included. It was necessary to adapt to the Armenian context. CCHF and anthrax were prioritized through a consultative participatory process in the COVID-19 era. Prioritization of pathogens rather than selection of zoonotic pathogens for One Health collaboration should be mentioned. There are some technical issues to be clarified;

Community veterinarians are paramedics or vaccinators? What is the qualification of community veterinarians?

Burning of carcasses infected with anthrax is not recommended due to biosecurity issues and burying is recommended. The paper should take into consideration of burying not burning of carcasses for the prevention and control of anthrax accordingly. English language needs to be improved and it will be good to make it concise.  

Comments on the Quality of English Language

There are typo errors and some sentences and expressions are difficult to understand. The language part should be improved and demands due attention. 

Author Response

The authors have worked under difficult situations, i.e. Covid-19 pandemic and they tried to do their best to prepare this manuscript. The importance of OH and challenges in One Health integration of different sectors for prevention and preparedness for VBDs and zoonoses has been elaborated in the Armenian context. The proposed One Health Conceptual Framework is based on the impact of climate change on the emergence of vector-borne diseases of zoonotic origin. The authors have recognized the limitations of the study. It will be good to reflect on the Joint External Evaluation report and recommendations to be linked to strengthening IHR capacities such as preparedness, risk assessment, response, and multisectoral coordination  (One Health) and surveillance rather than IHR (2005) itself.

Stakeholder mapping (not analysis) to analyze roles and responsibilities of different sectors in prevention and preparedness for VBDs and zoonoses. MLS project was designed to understand the One Health approach for vector-borne diseases such as CCHF, RVF, and WNV in the context of climate change in Africa, and Armenia was also included. It was necessary to adapt to the Armenian context. CCHF and anthrax were prioritized through a consultative participatory process in the COVID-19 era. Prioritization of pathogens rather than selection of zoonotic pathogens for One Health collaboration should be mentioned. There are some technical issues to be clarified;

Community veterinarians are paramedics or vaccinators? What is the qualification of community veterinarians?

Burning of carcasses infected with anthrax is not recommended due to biosecurity issues and burying is recommended. The paper should take into consideration of burying not burning of carcasses for the prevention and control of anthrax accordingly. English language needs to be improved and it will be good to make it concise.  

Comments on the Quality of English Language

There are typo errors and some sentences and expressions are difficult to understand. The language part should be improved and demands due attention. 

Submission Date

06 December 2023

Date of this review

21 Dec 2023 17:28:48

Authors’ reply

Dear reviewer,

thanks a lot for your comments which helped us to reflect and revise some areas of the manuscript.

In particular:

With reference to the point on IHR-2005 vs Joint External Evaluation: we appreciated your comment. The outcomes of previous assessments like the WHO Joint External Evaluation of IHR Core Capacities in 2016 [37] and  the National Bridging Workshop on IHR and PVS Pathway in 2019 [38] were analysed during the preliminary phase of the study and considered during the workshops. This has been added to the revised manuscript in Table 2 and in the conclusions (lines 460-469 of the revised manuscript).

Prioritization of pathogens rather than selection of zoonotic pathogens for One Health collaboration should be mentioned: we agree on this comment and we amended accordingly in the revised manuscript (chapter 3.1 and 3.2)

Community veterinarians are paramedics or vaccinators? What is the qualification of community veterinarians? : They are specialists who graduated from a veterinary higher education institution. We integrated this information in the revised manuscript (lines 343-348)

Burning of carcasses infected with anthrax is not recommended due to biosecurity issues and burying is recommended. The paper should take into consideration of burying not burning of carcasses for the prevention and control of anthrax accordingly: as you know, this topic is debated, and both WHO webcove97.5 Africa.doc (who.int) and WHOA guidance Anthrax - WOAH - World Organisation for Animal Health report incineration or alternatively  deep burial as disposal methods. In Armenia they  discussed on both burning and burying as disposal methods in accordance with specific situations and feasibility.

We amended the sentence (line 420)

English language needs to be improved and it will be good to make it concise: we revised the entire manuscript as per your suggestion.

Kindly see in attachment the revised manuscript
